# Cultural Perception of the Historical and Cultural Blocks of Beijing Based on Weibo Photos

**Siyu Chen** [1,2], **Bin Meng** [1,2,*], **Na Liu** [1,2,3], **Zhenyu Qi** [1,2,4], **Jian Liu** [2,5] **and Juan Wang** [1,2]

1    College of Applied Arts and Science, Beijing Union University, Beijing 100191, China;
     20201070510119@buu.edu.cn (S.C.); 20211070510105@buu.edu.cn (N.L.); qizhenyu@momenta.ai (Z.Q.);
     wangjuan@buu.edu.cn (J.W.)
2    Laboratory of Urban Cultural Sensing & Computing, Beijing Union University, Beijing 100191, China;
     liujian@cnu.edu.cn
3    Office of Party History Data Collection Committee of Yumen Municipal Committee of the Communist Party
     of China, Yumen 735200, China
4    Beijing Momenta Technology Co., Ltd., Beijing 100083, China
5    College of Resource Environment and Tourism, Capital Normal University, Beijing 100048, China
*    Correspondence: mengbin@buu.edu.cn

**Abstract:** Historic blocks are steeped in the history and culture of a city, reflecting the characteristics of the landscape during historical periods; they are of great significance to the preservation of the historical memory of the city. However, existing research generally lacks integration with big data, and research perspectives are mostly from the conservation planning of historic blocks, the evaluation mechanisms of blocks, and the development of block tourism resources; thus, the public perception is lacking. This study involved 28 historical and cultural blocks in Beijing, the capital of China, and constructed a system of cultural perception symbols based on the cultural connotations of the capital. On this basis, photo recognition was conducted on Weibo photo data collected by relying on the photo segmentation method, and the intrinsic factors affecting the cultural perception of the historical and cultural blocks were explored from the perspective of public perception (local residents and tourists). The results show that: (1) the capital culture of Beijing can be roughly divided into five categories: ancient capital culture, red culture, Beijing-style culture, innovation culture, and other types of culture, based on the photo recognition method; (2) from the perspective of public perception, the degree of perception of Beijing culture and innovation culture is generally higher, while the degree of perception of red culture and ancient capital culture is lower; (3) the 28 historic and cultural blocks of Beijing's old city are mainly dominated by one or more cultures, and there are no historic and cultural blocks with a balance of five cultures, reflecting the relative lack of cultural diversity within the blocks; (4) the local cultural identity of Beijing is prominent and dominant in the public mind. However, local residents have a relatively high perception of innovative culture, whereas tourists have a higher perception of ancient capital culture and red culture. In this study, photo recognition technology was introduced to study the cultural perception of historical and cultural blocks to provide new ideas and methods for the study of historical and cultural blocks.

**Keywords:** historical and cultural blocks; cultural perception; photo recognition; Weibo photo; culture types

## 1. Introduction

With the development of history, mankind has left behind many precious spiritual treasures, including cultural heritage. The value of cultural heritage is reflected in various aspects, such as history, culture, and art. As an ancient civilization, China has a unique and rich historical and cultural heritage. As one type of urban cultural heritage, a historical and cultural block is of great significance in preserving the historical memory of a city. It can reflect the characteristics of the historical period of a city and preserve its history

and culture. Given its cultural, scientific, educational, and aesthetic value, a historic and cultural block has very important conservation and research properties.

The historic quarter plays an important role in the cultural heritage of a city and is of great significance in preserving the historical memory of a city. It is steeped in the culture of the city as a whole. Kevin Lynch's 'Five Elements Theory' [1] formed the basis for studying the physical spatial structure of cities, focusing only on the physical aspects of the city and ignoring the social and cultural aspects. However, in his subsequent work *Good City Form* [2], Kevin Lynch further investigated the socio-cultural aspects of urban space to fill that gap. With the development of urbanization, it is important to preserve the physical objects in the city, but also to focus on the more difficult to analyze cultural aspects of them [3,4]. Therefore, from the perspective of enriching cultural heritage research, this paper focuses mainly on the cultural perception of historical and cultural blocks.

In recent years, the conservation and study of historical and cultural quarters have gained recognition and active participation from professional scholars, as they are a special cultural heritage with a dual identity, representing a thousand years of historical and cultural accumulation, as well as a space for the daily activities of residents. In 1996, the book *Revitalizing Historic Urban Quarters* summarized the history and development of historical and cultural quarters in Europe and North America and described various methods and approaches for the conservation and preservation of ancient historical and cultural quarters [5]. Since then, the protection and research of historical and cultural blocks have attracted the attention of many disciplines, such as geography, cultural heritage, tourism, urban planning, architecture, and so on [6–9]. At the same time, many new models and methods have emerged to study historical and cultural blocks: for example, the model used to measure the sustainability level of historical blocks, so as to realize the revitalization and sustainability of historical blocks [10]. Based on a comprehensive literature survey method, the model was built to assist the understanding of the revitalization principle of historic blocks [11].

In addition, most studies of historical and cultural blocks have focused on the preservation of cultural connotations [12–14], and it is not easy to reach consensus on the understanding and perception of cultural connotations [15,16]; even fewer studies have been conducted from the perspective of the public perception of cultural heritage. Moreover, most previous studies have focused on qualitative research [17] and a few studies have used questionnaires [18]. Therefore, this paper proposes a new research method that bridges two gaps: (1) compared to traditional questionnaires, big data, with its extensive public participation, compensates for the small sample size and high cost of traditional methods; (2) the technique of deep learning is invoked in the research field of historical and cultural heritage conservation, which promotes the renewal of research methods for historical and cultural heritage conservation in the information age. In addition, the combination of GIS spatial analysis methods also plays an important role in the comprehensive understanding of the cultural connotations of historic districts.

The historical and cultural block is a very special and important item of cultural heritage, carrying the historical and cultural accumulation of the old city of Beijing, showing the cultural connotation of the core area of the capital, and playing an indispensable role. Based on the definition of capital culture, this study builds a system of cultural symbols of perception using Weibo photos as the data source. Furthermore, with the aid of an photo recognition method using deep learning and a GIS spatial analysis method, from the perspective of the public perception of each block culture, this study also explores the present public awareness of historical and cultural blocks of Beijing, to allow the study of the conservation and utilization of historical and cultural blocks.

## 2. Review

### 2.1. Study of the Conservation and Development of Historic and Cultural Blocks

Research on the conservation planning related to historic and cultural blocks has focused on the protection of the block environment as well as policies and institutions

associated with the blocks. Historic blocks should be closely integrated into the landscape, focusing on the development of urban landscapes [19]. It is also worth noting that the shortcomings of the blocks should be complemented, with the emphasis on highlighting their characteristics and harmonizing them with their surroundings, and the focus on the conservation of cultural relics with irreversible damage and strengthening environmental protection in the blocks [20].

In terms of the evaluation mechanisms for historic and cultural blocks, when constructing evaluation mechanisms, not only should spatial elements, such as roads and buildings, in historic blocks be introduced into the evaluation system, but the interests of the original inhabitants should also be considered [21]. After World War II, socio-economic progress was made and tourism began to flourish, with most tourists preferring to visit culturally rich historical sites (historical buildings, monuments, etc.) rather than modern attractions [22]. The 'real' nature of a historic site is also one of the key reasons for its popularity with visitors [23]. The tourism development of historical and cultural blocks also plays an important role in the economic development of the region [24] and needs to be considered when constructing the evaluation system. It is not easy for policy makers to understand the needs and desires of historic block residents. Most studies on residential satisfaction focus only on social sustainability aspects and do not recognize the significance of increasing the satisfaction of residents in historic blocks. Therefore, this gap can be filled by assessing the perceptions of residents and non-residents regarding the importance of determinants of resident satisfaction in historic blocks [25].

### 2.2. The Study of Cultural Perception at the Traditional Macro Level

The concept of perception has its origins in psychological research; in this context, perception represents a process of mental activity that occurs when an external object has a visual or mental impact on an individual. The term cultural perception is based on the concept of psychological perception, in which external objects become cultural vehicles or cultural atmospheres that stimulate the individual, and the body processes the perceived cultural information through its organs to form a cultural understanding and impression of the area in which it is located.

Participants with different perceptions have different cultural perceptions. An increasing number of people are aware that the experience of smell can strongly influence their perception; by comparing the responses of Japanese and German to different everyday smells, a particularly pronounced difference was observed between the two groups in the level of pleasure, and significant differences were also found in the intensity scale of some odorants [26]. Differences in factors, such as culture, gender, beliefs, opinions, and human risk in different countries, can also lead to differences in cultural perceptions [27]. In addition, the mastery of temporal information can lead to differences in cultural perceptions. Research has shown that Chinese people are more aware of past events than Canadians, suggesting that Chinese people have a greater understanding of the past [28]. Tourists are the main body of the urban foreign population, and the cultural perception of tourists is the main component of urban cultural perception. The cultural perception of tourists refers to their experiences and psychological view of local culture when they visit a tourist destination. "Perceived attractiveness" reflects the emotions and opinions of an individual regarding the ability of a destination to meet their various needs [29]. The ultimate judgment of a destination determines the perceived attractiveness [30]. Both tourists and local residents are important subjects of the perception of urban culture, but the two groups are not well distinguished. In the era of big data, new methods have been updated to make the distinction between the two groups easy [31–33].

### 2.3. Research on Photo Recognition Technology with Rich Data Sources for Cultural Heritage

With the application and popularization of the internet, mankind has entered an era of big information explosion. The emergence of big data has significantly reduced the cost of data acquisition. Big data have the characteristics of "5Vs", namely volume,

variety, value, velocity, and veracity [34], which are properties that traditional data do not have; these confer big data with great potential in cultural perception research and make it a favored data source in academic research. In recent years, machine learning technology, which is widely used, has become increasingly mature, and photo data from social media has long been popular in the cultural research field because of its intuitiveness and wide range of data sources (e.g., features such as Flickr check-in data [35], street view photo photos [36], Panoramio photo data [37], remote sensing photos [38]) can more easily convey the perception and understanding of the surrounding environment to the outside world than text data. These data, aided by machine learning methods, and deep learning technology, have been increasingly used to protect cultural heritage in recent years. The Delft University of Technology used a convolutional neural network (CNN) combined with multiresolution imaging technology to identify that the automatic reconstruction of Vincent Van Gogh's paintings deteriorated over time [39]. Social media data can also be used to detect damage to cultural heritage, using convolutional neural networks to significantly reduce the amount of manual work required to find photographs of damaged cultural heritage [40]. Using the photo-sharing platform Flickr, the US Library of Congress has created a database of photos of cultural heritage, enabling users to view old photographs at any time [41]. Using a region-based photo segmentation framework, pictographs can be automatically extracted from degraded ancient Mayan codex photos, and these automatically extracted pictographs are comparable to retrieval results obtained using manual recognition [42].

By combing through the existing literature, it can be found that there is a wealth of research on historical and cultural districts, with research mainly focusing on the conservation planning, value evaluation mechanisms, and tourism development experiences of historical and cultural districts. Although the exploration of historical and cultural connotations is a research focus, the form of research is however mostly qualitative, and the research method is empirical research using traditional questionnaires, which has the problems of small sample size and high cost. In terms of research on cultural perceptions, researchers have focused more on macro-level and large-scale studies, and less on the cultural perceptions of cultural heritage of typical significance, such as historical and cultural blocks. At the same time, in the era of big data, the constant updating of technology has facilitated the development of cultural heritage conservation research, and more and more scholars have started to use photo recognition techniques (e.g., CNN), for the study of cultural heritage. Using the new techniques offered by deep learning, this paper investigates the cultural identity of historical and cultural blocks, refining cultural perceptions from the macro level to the micro and small scale. Because of these shortcomings in existing research, this paper seeks to make use of social media-based big data to expand the research community from a small questionnaire to as large a group of the public as possible, compensating for the small sample size and high costs of traditional methods.

In the remainder of this paper, we first introduce the data (Section 3.1) and methods for the historical and culture blocks: Cultural Symbol Recognition System (Section 3.2.1), Photo Classification (Section 3.2.2), Grouping Analysis (Section 3.2.3). Secondly, we examine the cultural perception of the historic and cultural district through four aspects: Number of Cultural Types within the Blocks (Section 4.1), Differences in Cultural Types within Blocks (Section 4.2), Cultural Combination of Different Blocks (Section 4.3), Differences in Cultural Perception of Different Types of Users (Section 4.4).

## 3. Materials and Methods

### 3.1. Materials

#### 3.1.1. Study Area

Beijing has been a city for more than 3000 years, and its capital has been established for more than 800 years. It has resulted in numerous historical relics and has a deep historical and cultural heritage in its long history. It is a modern metropolis with a strong ancient capital style, as well as being one of the first national historic and cultural cities in

China. The old city of Beijing is one of the richest areas in terms of historical and cultural resources, with historic and cultural blocks, character blocks, and other traditional blocks accounting for 40% of the total blocks [43]; it is the most important component of the conservation of the historic and cultural blocks of Beijing. Historic and cultural blocks are characteristic areas that embody traditional life and present the traditional appearance of the city. They constitute the most important substrate for the overall spatial characteristics of the old city of Beijing, which is an important area for the overall conservation of the city. Therefore, based on *The Detailed Control Plan for the Core Area of the Capital Function (Block Level) (2018–2035)* [44] and with reference to the *Planning Commission. Conservation Planning of 25 Historic Areas in Beijing Old City in 2002* [45], as well as the *Beijing Urban Master Plan (2016–2035)* [46], the research object of this study was the 28 historical and cultural blocks in the old city of Beijing. (Figure 1)

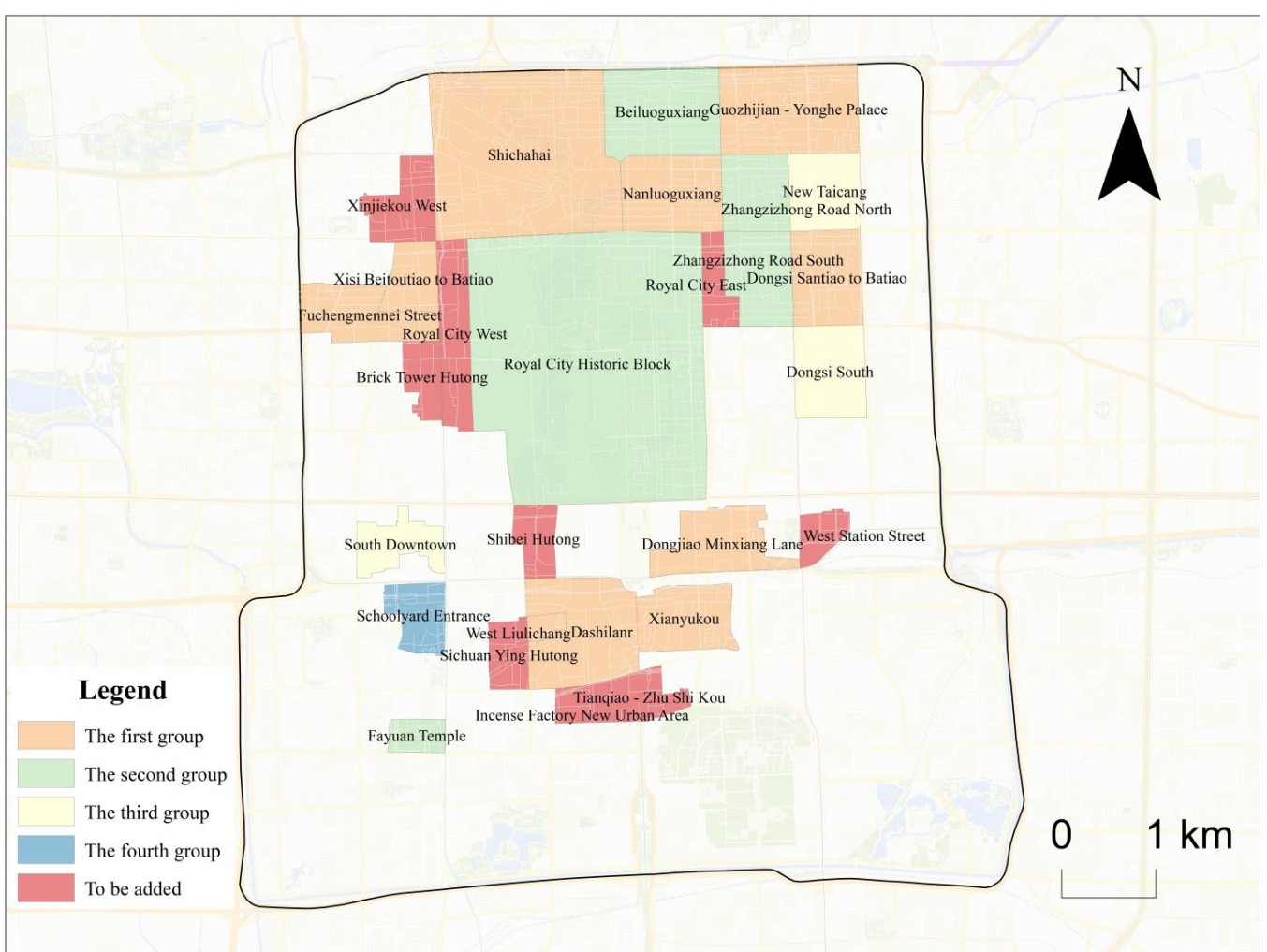

**Figure 1.** Spatial distribution of historic and cultural blocks.

### 3.1.2. Research Data

With the help of the Sina Weibo API and web crawler tools to obtain Weibo photo data within the old city of Beijing in 2019, the data elements included information, such as user ID, photo path, check-in latitude and longitude, yielding a total of 1,119,000 Weibo users and 4,324,000 Weibo photos, each with a resolution of 96 dpi × 96 dpi. According to the Weibo Data Centre (http://data.weibo.com/datacenter/recommendapp, accessed on 30 September 2020), the Weibo User Trend Report 2020 showed that the number of monthly active Weibo users increased to 511 million as of September 2020, with an average

of 224 million daily active users. These figures indicate further consolidation of Sina Weibo as the leading social media platform in China. However, we are also aware of the problems of sample bias and limited representativeness of social media data. Specifically, social media platforms are used by a relatively young group of people, and the users of social media may vary according to socioeconomic attributes (such as age, gender, and occupation) and individual behavioral differences. This study takes advantage of a large amount of information in Weibo data, which can help us to comprehensively analyze the differences in public perceptions of culture in the historic and cultural blocks of the old city of Beijing. This study breaks the information delay problem of previous questionnaire data and strengthens the effectiveness and relevance of policy recommendations for the conservation and utilization of the historic and cultural blocks of Beijing's old city.

*3.2. Methodology*

This study examines the cultural perceptions of 28 historical and cultural blocks in Beijing's old city. First, a cultural perception symbol recognition system was constructed based on the cultural connotation of the capital city. The micro-blog photos were then combined with the cultural symbol recognition system for manual classification. After manual recognition, a photo recognition training model based on cultural classification was established. The accuracy of the model reached 81%, based on an accuracy test. Second, these five types of cultural photos were presented in a geographical space according to the coordinates of the photo. Finally, the results were analyzed based on four aspects: the number of cultural types in the block, the difference in cultural types in the block, the cultural combination in different blocks, and the difference in cultural perception among users of different types. The research framework is illustrated in Figure 2.

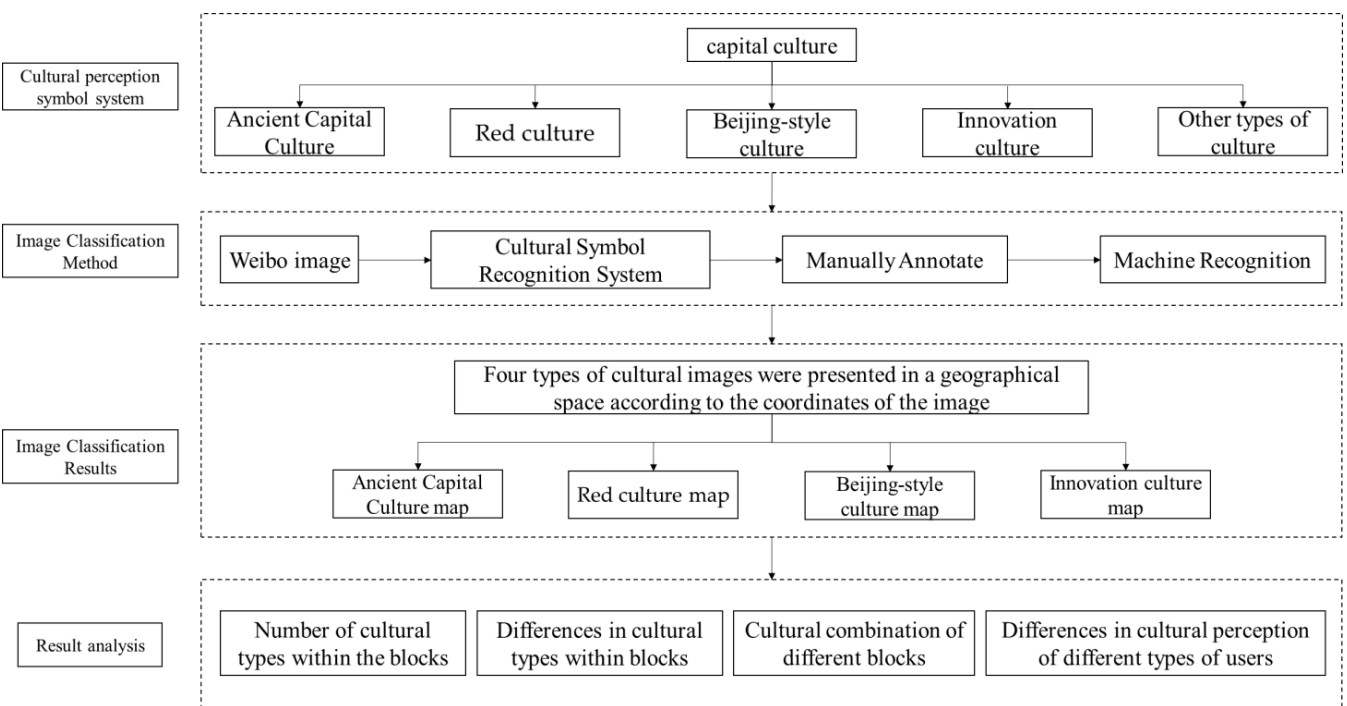

**Figure 2.** Study framework.

3.2.1. Cultural Symbol Recognition System

Culture is an abstract and complex concept involving relativity. Therefore, in the process of sorting cultural perception symbols, this study found that the historical and cultural blocks include four types [47]: ancient capital culture with a long history; rich and thick red culture; distinctive Beijing-style culture; and booming innovative culture. There are also typical systematic cultures that are not covered by these four

types of culture (such as afternoon tea culture and internet celebrity culture), which are collectively referred to as other types of culture in this paper. Because these types of cultures do not belong to the official definition of capital culture, a specific analysis was not made in this study. Capital culture and Beijing culture are both inclusive concepts in China, but they have clear emphases: the former emphasizes national representativeness based on region, while the latter emphasizes regional characteristics. Capital culture covers cultural resources and cultural phenomena in Beijing, but delineates the narrow limitations of the region [48].

Ancient capital culture comprises the artifacts, institutions, and spiritual forms of culture that have been formed during the long history of Beijing as a capital city, and it is the most representative cultural form of the excellent traditional culture of the city. Ancient capital culture is the interplay of the regional and functional culture of Beijing, and it is the typical mark of the culture of the city and the root of its cultural lineage.

Red culture is an advanced culture with the Communist Party of China at its core, created by the Chinese Communist Party during the revolution, construction, and reform processes. Red culture is the core connotation, valuable resource, and the soul of the capital's culture.

Beijing-style culture is the characteristic culture of Beijing, and it is a culture with a strong life atmosphere formed during the long historical development process of Beijing. It is embodied by the folk and language cultures of Beijing, such as traditional etiquette, food, dialect, siheyuan, and hutong cultures. Beijing flavor culture includes not only the unique cultural traditions of Beijing, but also the subtle shaping of the psychology and humanistic qualities of the people in Beijing by these cultures. Beijing-style culture has obvious regional characteristics and is the verve of capital culture.

Innovation culture is a cultural form in which the atmosphere of innovation and innovative thinking penetrate into various fields of economic and social development, including culture. It is a kind of innovation atmosphere and innovation consciousness of the entire nation. Innovation is the lasting driving force of development in Beijing, and the culture of innovation has contributed to the cultural construction of the capital city.

Based on the connotative characteristics of the capital's culture, this study introduced its classification criteria and combined the cultural resources within the 28 historical and cultural blocks, as well as the content of the Weibo sign-in pictures, to sort out and summarize the different types of cultural perception symbols and build a reasonable system of cultural perception symbols (Table 1).

**Table 1.** Cultural perception symbol system.

| Cultural Types | Symbol Category | Concrete Symbol Example | Judgment Criteria | Picture Sample |
|---|---|---|---|---|
| Ancient Capital Culture | The city palace | The Forbidden City Complex, the Prince's Mansion | If it is material culture and spiritual culture related to the ancient capital. |  |
| | Temple garden Provincial temples Road transport | The Temple of Heaven, the Taimiao Shuntianfu Hall, Catholic Church Wanning Bridge, Yinding Bridge | | |
| | River lake system Architectural ornament Religious culture The national etiquette Idea | Moat, Grand Canal Ridge beast, Huabao Royal religious rites The layout of the palace building Zhou Li examination of labor records | | |

**Table 1.** *Cont.*

| Cultural Types | Symbol Category | Concrete Symbol Example | Judgment Criteria | Picture Sample |
|---|---|---|---|---|
| Red culture | Remains of historic buildings | Former residences of leaders and other progressive figures, meeting sites | If it is a material and spiritual culture that is co-created by the Chinese Communists, advanced elements, and the masses and if it contains a rich revolutionary spirit. |  |
| | Modern architecture | Memorial Hall, Exhibition Hall, Monument, Martyr's Cemetery, Memorial Square, Memorial Sculpture | | |
| | Movable appliance items | Flags, books, newspapers, slogans, couplets, poems, inscriptions, inscriptions, weapons, and ammunition, etc. | | |
| | Social individual | Li Dazhao, Chen Duxiu, Song Qingling | | |
| | Collection of individuals | Haobalian on Nanjing Road, Prairie Hero Sisters | | |
| | Important and positive behaviors | Creation, Commendation, Conference, Uprising | | |
| | Widely recognized spirit | The spirit of the May Fourth Movement, the spirit of the Beijing Olympics, the spirit of the Great Northern Wilderness | | |
| | Remains of historic buildings | Former residences of leaders and other progressive figures, meeting sites | | |
| | Modern architecture | Memorial Hall, Exhibition Hall, Monument, Martyr's Cemetery, Memorial Square, Memorial Sculpture | | |
| | Movable appliance items | Flags, books, newspapers, slogans, couplets, poems, inscriptions, inscriptions, weapons and ammunition, etc. | | |
| | Social individual | Li Dazhao, Chen Duxiu, Song Qingling | | |
| | Collection of individuals | Haobalian on Nanjing Road, Prairie Hero Sisters | | |
| | Important and positive behaviors | Creation, Commendation, Conference, Uprising | | |
| | Widely recognized spirit | The spirit of the May Fourth Movement, the spirit of the Beijing Olympics, the spirit of the Great Northern Wilderness | | |
| Beijing-style culture | Clothes | Manchu color cheongsam | If it is a material culture and spiritual culture that is dominated by civilian culture, integrates diversity, is rooted in the people, and is directly related to public life. |  |
| | Food | Beijing-style snacks: fried liver, bean juice, enema | | |
| | Housing | Siheyuan, Hutong, compound | | |
| | Transportation | Tramways, rickshaws | | |
| | Temple fair | Burning incense for blessing, visiting Baita Temple, offering sacrifice to the God of Wealth | | |
| | Folk custom | Han marriage customs "six rites", Manchu marriage customs, etc. | | |
| | Consuming behavior | Stores, commercial areas | | |
| | Entertainment behavior | Flower party, craftsmanship, stage, teahouse | | |
| | Folk handicrafts | Sugar painting, hairy monkey, carved lacquer, Beijing embroidery, cloisonne | | |
| | Beijing-style literature | Prose, poetry, fiction | | |
| | Beijing-style film and television | Movies, TV dramas, plays | | |
| | Beijing-style opera | Beijing Opera, Beijing Cross talk | | |

**Table 1.** *Cont.*

| Cultural Types | Symbol Category | Concrete Symbol Example | Judgment Criteria | Picture Sample |
|---|---|---|---|---|
| Innovation culture | Genre, School | Jiagan School, Wuxu movement, new Culture Movement | If it is material culture and spiritual culture that reflects the bold exploration and creative spirit of the people. |  |
| | Cultural Works | Stone Buddhist scriptures, Yuan Qu, new school | | |
| | Technological Achievements | Tonghui River, Shoushi Calendar, Beijing–Zhangjiakou Railway | | |
| | Innovation environment | Colleges and universities, libraries, exhibition halls, scientific research institutes | | |
| | Innovative talents | High-level talents, entrepreneurial youth, angel investors | | |
| | Genre, School | Jiagan School, Wuxu movement, new Culture movement | | |
| | Cultural Works | Stone Buddhist scriptures, Yuan Qu, new school | | |
| | Technological Achievements | Tonghui River, Shoushi Calendar, Beijing-Zhangjiakou Railway | | |
| | Innovation environment | Colleges and universities, libraries, exhibition halls, scientific research institutes | | |
| | Innovative talents | High-level talents, entrepreneurial youth, angel investors | | |
| Other types of culture | Foreign culture | Festivals, meals, musical instruments | Typical cultures that exist in historical and cultural blocks but do not belong to the cultural type of the capital. |  |
| | Sports Culture | Basketball, soccer, yoga, fitness | | |
| | Natural landscape | Flower arrangement, potted plants, green landscape | | |
| | Internet pop culture | Forbidden City, Hanfu, internet celebrity shops, Internet celebrity snacks | | |

### 3.2.2. Photo Classification in Deep Learning

The concept of deep learning was proposed in 2006. It is a type of network structure with multiple hidden layers and multiple perceptions that can describe the attributes and characteristics of objects on a more abstract and deeper level [49]. Fast AI, founded by Jeremy Howard in 2017, is a deep learning library that provides high-level components that can quickly and easily deliver state-of-the-art results in standard deep learning domains and lower-level components that can be combined by researchers to create new methods [50]. This is possible owing to the layered architecture of Fast AI, which expresses the common underlying patterns of many deep learning and data processing technologies in decoupled abstractions. Fast AI can express these abstractions cleanly and concisely by taking advantage of the dynamic nature of the underlying Python language and the flexibility of the PyTorch library. It can use a few lines of code to build an photo classifier, photo segmentation model, text sentiment model, recommendation system, and table model [51].

Using Fast AI, this study classified and identified Weibo photo data from social media data based on the cultural connotations of historical and cultural blocks, as well as the cultural connotations of the capital's culture. This allows a more comprehensive and accurate grasp of the degree of public perception of the cultural types of each historical and cultural block and provides a new research perspective for the conservation and utilization of historical and cultural blocks.

### 3.2.3. Grouping Analysis

Spatial analysis methods primarily use grouping analysis methods for spatial clustering. The grouping analysis method is mainly used to find natural clusters that exist in the data and have certain similar attributes to perform the classification. Given the number of groups to be created, it looks for a solution that will make all the elements in each group as similar as possible, but as different as possible from one group to another. Element similarity is based on a set of properties specified for the parameters of the analysis

field and can also include spatial or spatial-temporal properties [52,53]. This study used the grouping analysis tool of ArcGIS to classify historic and cultural blocks with similar attributes based on the two attribute fields of "total number of cultures" and "percentage of each type of culture". This can facilitate the understanding and the analysis of the intrinsic reasons for the formation of cultural types in the 28 historic and cultural blocks of Beijing's old city in order to promote the conservation and development of historic and cultural blocks as a whole.

## 4. Results

### 4.1. Number of Cultural Types within the Blocks

A total of 1.897 million Weibo photos were classified and identified by deep learning methods: 211,000 photos were identified in the category of ancient capital culture perception, accounting for 11.13%; 177,000 photos in the category of red culture perception, accounting for 9.3%; 492,000 photos in the category of Beijing-style culture perception, accounting for 26%; 326,000 photos in the category of innovation culture perception, accounting for 17.2%; 226,000 photos of other types of cultural perception, accounting for 11.93%; and 465,000 photos unrelated to cultural perception, accounting for 24.49%. After removing photos unrelated to cultural perception, the total number of photos in the five categories of cultural perception was 1,432,000. Table 2 presents the number of cultural types perceived within each historical and cultural block.

**Table 2.** Statistics on the number of cultures in the 28 historical and cultural blocks of Beijing's old city.

| Type of Culture / Block Name | Ancient Capital Culture | Red Culture | Beijing-Style Culture | Innovation Culture | Other Types of Cultural | Five Types of Culture |
|---|---|---|---|---|---|---|
| Royal City Historic Block | 157,649 | 107,834 | 127,337 | 95,362 | 74,824 | 563,007 |
| South Downtown | 437 | 724 | 3880 | 5588 | 1570 | 12,199 |
| Fayuan Temple | 967 | 399 | 3621 | 1119 | 2883 | 8989 |
| Dongjiao Minxiang Lane | 4610 | 8565 | 20,476 | 23,117 | 13,133 | 69,908 |
| Schoolyard Entrance | 170 | 322 | 1239 | 1187 | 824 | 3742 |
| Xisi Beitoutiao to Batiao | 228 | 314 | 1532 | 825 | 708 | 3607 |
| Brick Tower Hutong | 236 | 500 | 2269 | 3731 | 1175 | 7911 |
| Fuchengmennei Street | 1440 | 1375 | 5369 | 3710 | 2945 | 14,839 |
| Royal City West | 626 | 862 | 5250 | 3374 | 2336 | 12,448 |
| Nanluoguxiang | 4323 | 5782 | 52,760 | 25,891 | 18,661 | 107,417 |
| Beiluoguxiang | 1860 | 2003 | 14,430 | 8417 | 7125 | 33,836 |
| Xinjiekou West | 433 | 855 | 4720 | 2361 | 2027 | 10,396 |
| Shichahai | 12,417 | 12,171 | 70,758 | 41,892 | 23,549 | 160,787 |
| Royal City East | 1882 | 5079 | 8964 | 8755 | 6144 | 30,824 |
| Zhangzizhong Road North | 384 | 750 | 5627 | 3278 | 2613 | 12,652 |
| Zhangzizhong Road South | 1436 | 2796 | 8505 | 10,835 | 4383 | 27,956 |
| Guozhijian—Yonghe Palace | 11,306 | 6758 | 49,303 | 22,243 | 22,068 | 111,678 |
| New Taicang | 589 | 1144 | 9334 | 6297 | 5519 | 22,883 |
| Dongsi South | 807 | 1356 | 6353 | 4843 | 3296 | 16,655 |
| Dongsi Santiao to Batiao | 672 | 989 | 6279 | 3851 | 3152 | 14,943 |
| Shibei Hutong | 144 | 331 | 801 | 754 | 499 | 2529 |
| Xianyukou | 1123 | 1627 | 10,198 | 4816 | 3620 | 21,384 |
| West Liulichang | 112 | 250 | 1664 | 780 | 388 | 3194 |
| Sichuan Ying Hutong | 106 | 267 | 1233 | 984 | 758 | 3348 |
| Dashilan | 6561 | 12,416 | 66,666 | 38,359 | 20,118 | 144,120 |

The cultural perceptions in the 28 historical and cultural blocks of the old city were classified into five levels: high, medium-high, medium, medium-low, and low. The results, as shown in Figure 3, indicate that there are significant spatial differences in the perceived cultural heat of different historical and cultural blocks, with areas such as the Royal City historic block, Shichahai, and Dashilan being higher than some areas away from the Forbidden City.

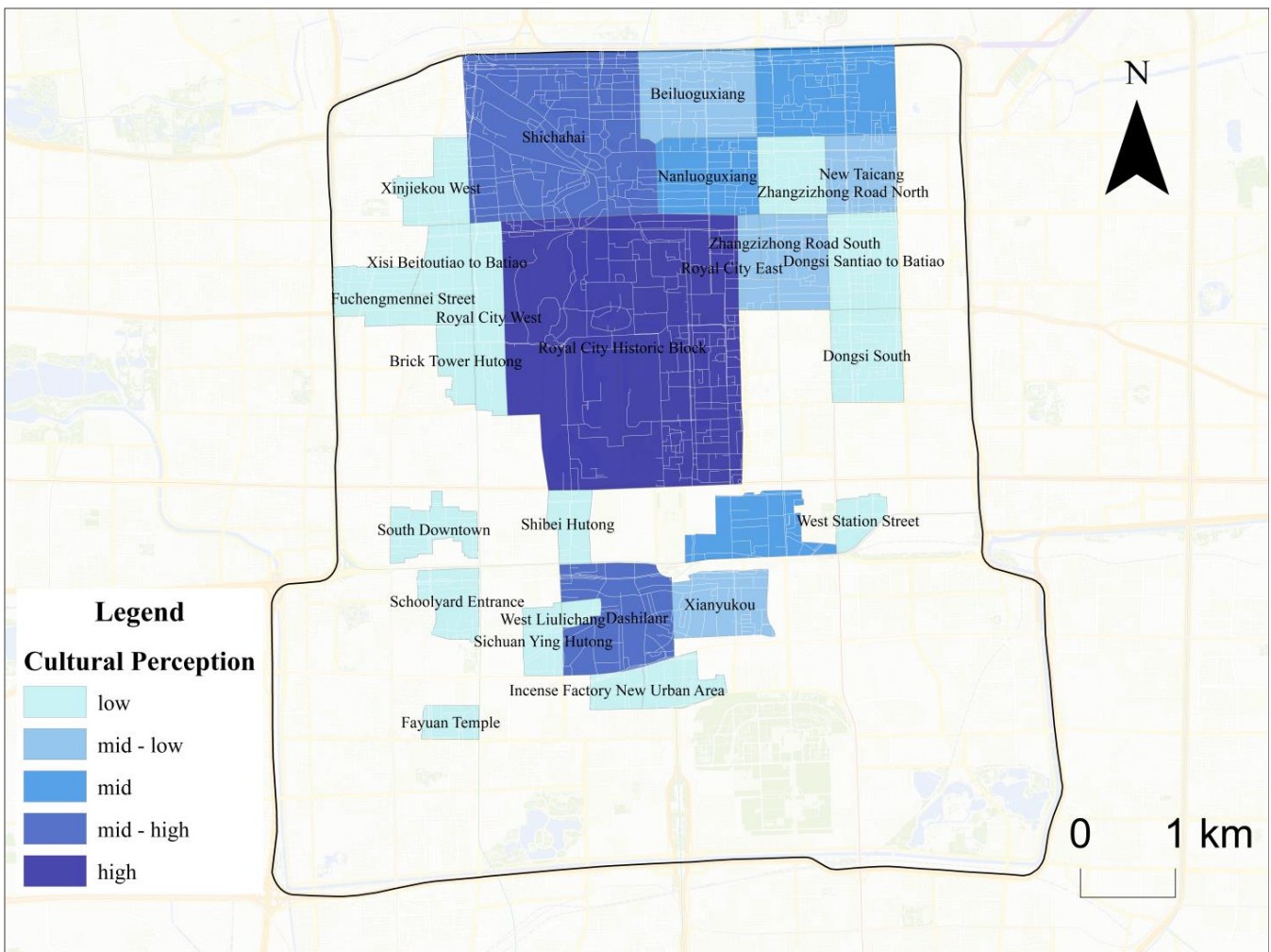

**Figure 3.** Cultural perception results for 28 historic and cultural blocks in Beijing's old city.

*4.2. Differences in Cultural Types within Blocks*

In order to further analyze the spatial differences in the perceived differences of cultural types in different historical and cultural blocks, this paper, using an ArcGIS natural interruption point grading method, presents the distribution characteristics of the absolute number and relative number (the ratio of the number of a certain type of culture to the total number of cultures in the block) of each of the five types of culture within the 28 historical and cultural blocks. Furthermore, it highlights the relative and absolute numbers of a certain type of culture in all the blocks. The relative and absolute amounts of a particular type of culture in all blocks are divided into five classes (Figure 4), with a, b, c, and d representing the perceptions of ancient capital culture, red culture, Beijing-style culture, and innovative culture, respectively. The first, second, and third columns represent the absolute number of cultural perceptions, relative number of cultural perceptions, and combination of cultural perception characteristics, respectively.

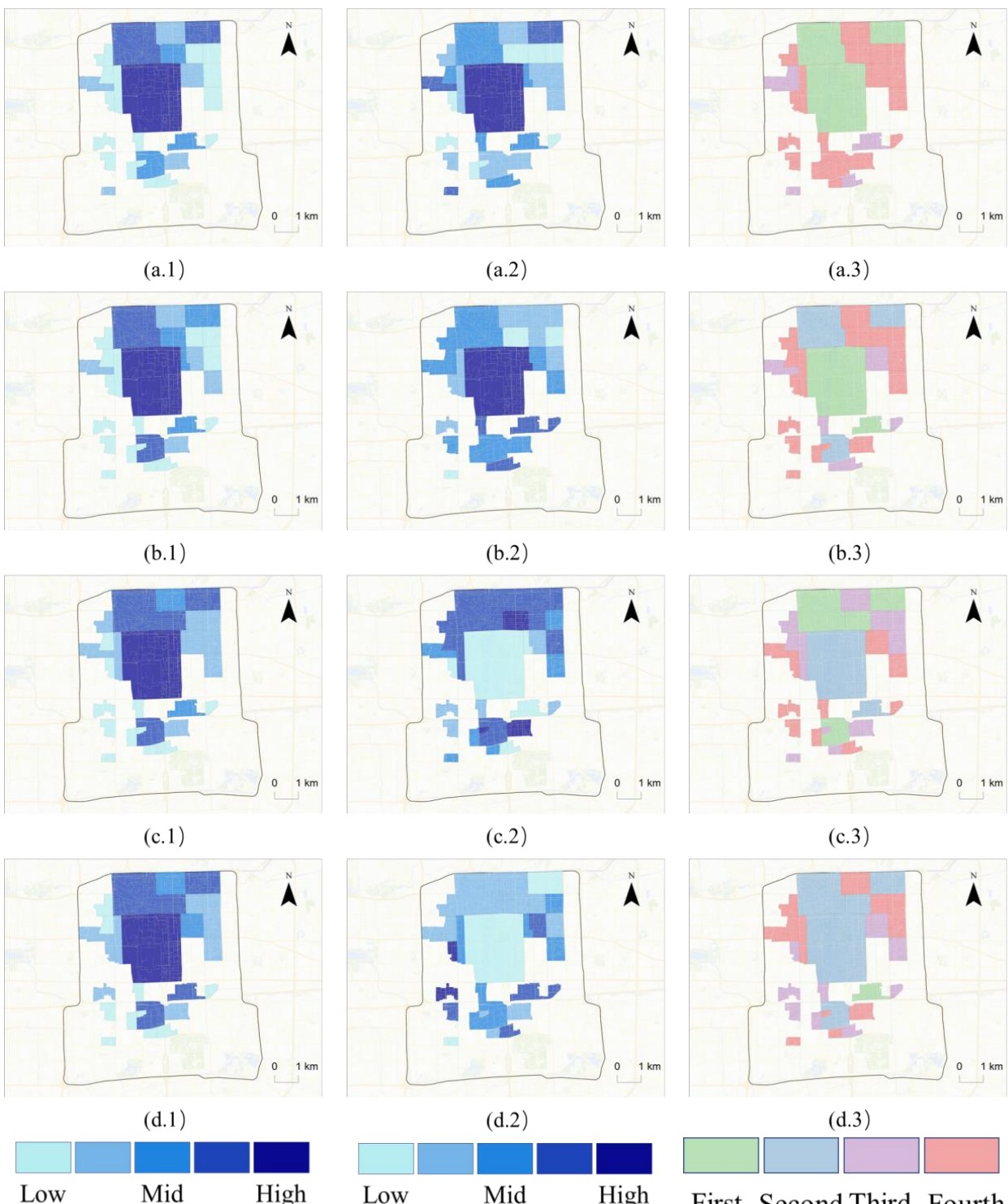

**Figure 4.** Distribution of absolute number, relative number, and four quadrants of various cultures in each block.

The distribution of the absolute and relative numbers of cultural perceptions of different blocks in Figure 4 reveals that, in absolute numbers (a.1-d.1), the Royal City historic block has higher cultural perceptions than other blocks. However, in relative numbers (a.2-d.2), the Royal City historic block has the highest perceptions in ancient capital culture and red culture; the Xiangyukou historical and cultural block, the West Liulichang historical and cultural block, and the Nanluoguxiang historical and cultural block have

higher perceptions of the Beijing-style culture than other blocks, and the South Downtown historical and cultural block and the Brick Tower Hutong historical and cultural block have the highest perceptions in innovation culture.

　　In addition, absolute and relative numbers were used to construct a four-quadrant graph to represent the combined characteristics of cultural perception. Taking Beijing-style culture as an example (Figure 5), Nanluoguxiang and the other three blocks in the first quadrant have absolute advantages in both the absolute and relative numbers of Beijing-style culture perceptions in Weibo, indicating that the Beijing-style culture in these three blocks has a prominent dominant position. In the second quadrant, the absolute number of Beijing-style culture perceptions in Weibo is high for Dongjiao Minxiang Lane and the Royal City Historic Block, but the relative number is low, which indicates that the public's perception of Beijing-style culture in this block is low, and the development status of this kind of culture in this block is relatively weak compared with those of the other four types of culture. In the third quadrant, the absolute number and relative number of Beijing-style culture perceptions in Weibo in nine blocks, such as the Royal City East, are low, indicating that the overall perception of Beijing-style culture in these blocks is poor. The absolute number of culture-aware Weibos is low, but the relative number is high in Xianyukou and in the other eleven blocks in the fourth quadrant, indicating that there is a strong atmosphere of this kind of Beijing-style culture in these blocks. The inheritance and preservation of the Beijing-style culture are relatively good, and the public has a high degree of its awareness

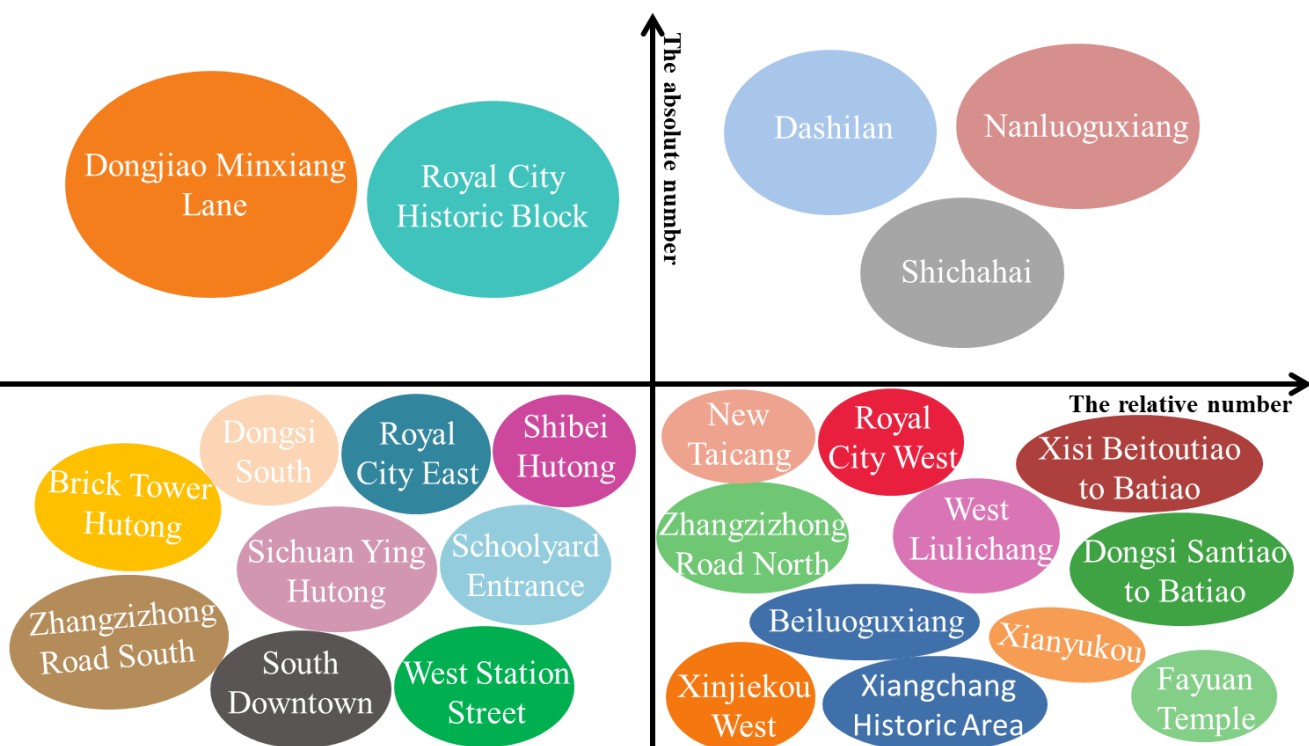

**Figure 5.** Four-quadrant classification of absolute and relative number of Beijing-style culture.

　　By comparing the changes in the absolute and relative amounts of culture in each block, the 28 historic and cultural blocks can be categorized into the following three variations: (1) the absolute and relative amounts are close, (2) the absolute amount is low and the relative amount is high, and (3) the absolute amount is high, but the relative amount is low.

　　It was observed that among the ancient capital cultures (Figure 4(a.3)), the Royal City Historic Block is in the first quadrant, with the absolute and relative number of perceptions of ancient capital culture being higher than those of the other 27 historic and cultural blocks. This indicates that the appearance of the ancient city was properly maintained, the cultural

atmosphere is strong, the current state of conservation and use in the block is consistent with its planning position, and the public perception of the culture is strong.

For red culture (Figure 4(b.3)), the Royal City East is in the middle of the five classes in absolute terms, but its relative quantity is the highest, belonging to the fourth quadrant, indicating a strong public perception of the red culture within the quarter. The block is geographically close to Beichizi Street, and representative red culture carriers, such as the Red Building of Peking University, the Beijing New Culture Movement Memorial Hall, and Qiushi Magazine, are only one street away from the block, making the atmosphere of red culture in the block relatively strong.

For Beijing-style culture (Figure 4(c.3)), the three historic blocks of Xiangyukou, West Liulichang, and Nanluoguxiang had the highest perception. These three blocks are the oldest historical and cultural blocks in Beijing, and their architectural and cultural connotations are properly preserved.

For innovation culture (Figure 4(d.3)), South Downtown and Brick Tower Hutong had the highest public perception. The South Downtown and Brick Tower Hutong have the highest level of perceived innovation culture, and it is possible that these two are the core windows for the display of high technology in the city of Beijing, which has contributed to the high level of public perception of innovation culture.

### 4.3. Cultural Combination of Different Blocks

In order to have a good understanding of the intrinsic association and spatial distribution characteristics of blocks of the same cultural type in 28 historical and cultural blocks, this study used the grouping analysis tool of ArcGIS to identify and classify historical and cultural blocks with similar cultural types, and the results are shown in Figure 6.

Based on the optimal assessment of the ArcGIS grouping analysis, the 28 historic and cultural blocks were divided into seven groups (Figure 6), and the classification characteristics of each group are shown in Table 3.

The cultural type of the Group I block is dominated by ancient capital and red cultures. This block is located at the core of the old city and has the largest area (27%) of the 28 historical and cultural blocks. The blocks dominated by ancient capital culture and red culture are linked to the actual situation, fully exploiting intrinsic cultural values and creating a scientific and reasonable development line tailored to the different pieces.

The cultural types in Group II are dominated by Beijing-style culture and other types of culture. The direction of the conservation and use of this group of blocks is based on the strengths of the blocks, with more exploration of the cultural connotations of the blocks and the preservation and inheritance of the Beijing flavor culture that is already present.

Group III is dominated by a culture of innovation. Although not spatially adjacent to each other, this group is surrounded by several large leisure and shopping venues, financial and commercial facilities, and other buildings with a strong modern atmosphere.

Group IV is dominated by Beijing-style culture and innovation culture. This group contains a large number of well-preserved hutong and courtyard buildings and uses a point-by-point approach to protect and utilize the surrounding areas, leading to the overall development of the block and presenting the connotations of Beijing-style culture in a comprehensive manner. The level of commercial development in these blocks is much higher than that in other blocks of the same planning and positioning, which has undoubtedly attracted many high-tech, cultural, and creative industries in the area, injecting innovative vitality into the area and, at the same time, allowing for the development of a culture of innovation.

Group V is a mixture of the three types of culture: Beijing-style, innovation, and other types of culture. In terms of spatial distribution characteristics, the blocks in this group are adjacent to one another, with the exception of the Royal City West and Sichuan Ying Hutong blocks. The blocks are all connected, and the cultural connotations of the blocks interpenetrate and intermingle, creating a diversity of cultural types in this group.

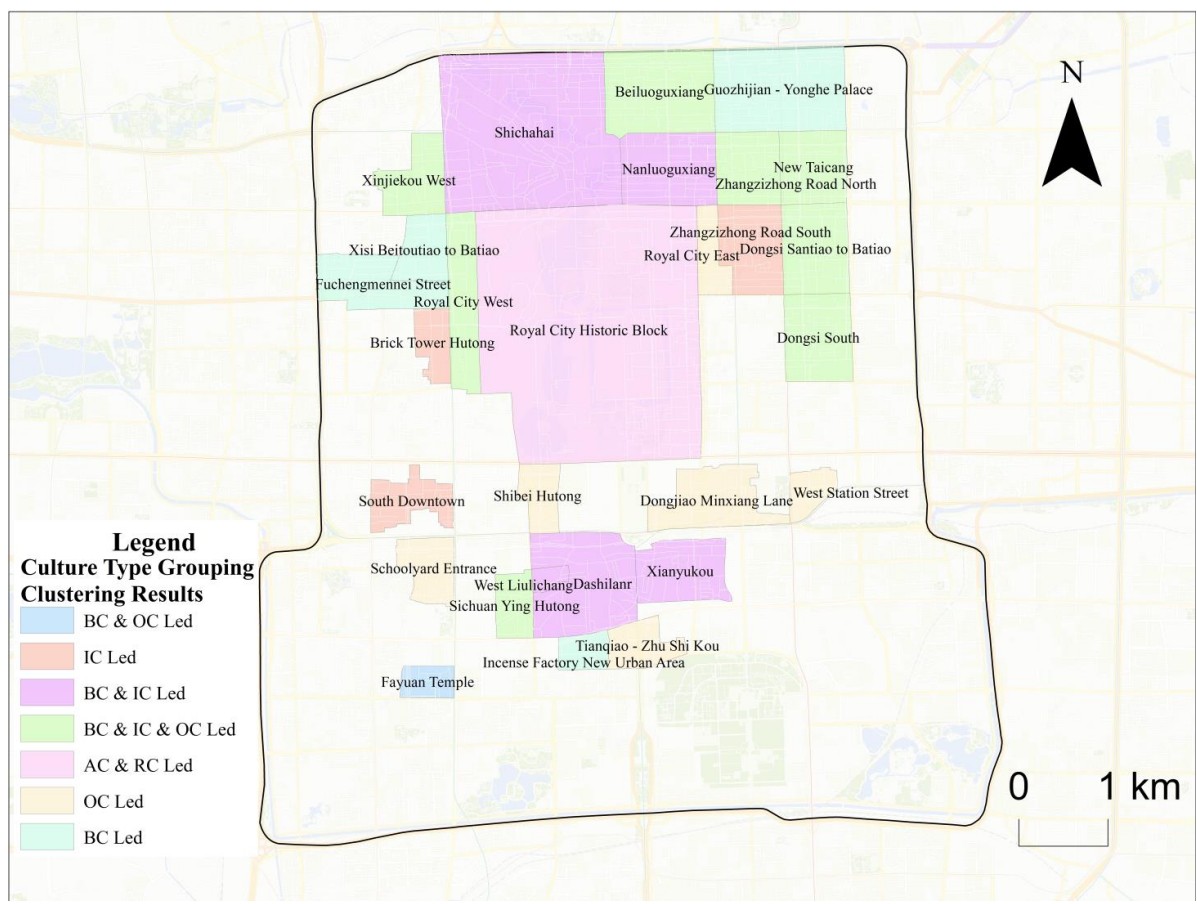

**Figure 6.** Clustering results of cultural types grouped in 28 historical and cultural blocks in the old city of Beijing.

**Table 3.** Grouping clustering characteristics.

| Group | Type of Culture | Block | Location of Distribution | Form of Distribution |
|---|---|---|---|---|
| I | Ancient capital culture, Red culture-led | Royal City Historic Block | Old City Centre | / |
| II | Beijing-style culture, Other types of culture-led | Fayuan Temple | Southwest of the old city | / |
| III | Innovation culture-led | Brick Tower Hutong, South Downtown, Zhangzizhong Road South | Along both sides of Chang'an Avenue, adjacent to large business blocks | Discrete distribution |
| IV | Beijing-style culture, Innovation culture-led | Shichahai, Nanluoguxiang, West Liulichang, Dashilan, Xianyukou | Along both sides of the central axis, distributed in the north and south directions of the old city. | Sheet distribution |
| V | Beijing-style culture, Innovative culture, and Other cultures are mixed and lead | Xinjiekou West, Royal City West, Beiluoguxiang, Zhangzizhong Road North, New Taicang, Dongsi Santiao to Batiao, Sichuan Ying Hutong, Dongsi South | Mostly located in the north of the old city | Sheet distribution |
| VI | Other types of culture-led | Royal City East, Shibei Hutong, Schoolyard Entrance, Dongjiao Minxiang Lane, West Station Street, Tianqiao—Zhu Shi Kou | Mostly located in the south of the old city | Discrete distribution |
| VII | Beijing-style culture-led | Xisi Beitoutiao to Batiao, Fuchengmennei Street, Guozhijian—Yonghe Palace, Incense Factory New Urban Area | Mostly located in the north of the old city | Discrete distribution |

Group VI is a group of blocks dominated by other types of cultures. The buildings in this group are mostly Western in style and maintain an exotic character in general, while hotels, religious buildings, and educational buildings also occupy a large number of blocks.

Group VII of the historic and cultural blocks is dominated by Beijing-style culture. The planning and positioning of this group of blocks are both a comprehensive conservation area for scenic spots and a traditional hutong residential conservation area. There are a large number of cultural relics and monuments, a large number of courtyards, and a largely intact street pattern, reflecting a certain extent the orthodox taste and official culture of the courtyard style of the inner-city streets and lanes and a high historical and cultural value.

*4.4. Differences in Cultural Perception of Different Types of Users*

Based on the posting frequency and time of Weibo users, we divided the Weibo data obtained into local residents and tourists, according to user attributes under the classification conditions of posting <800 posts a year and posting for three consecutive months. Furthermore, we conducted a comparative study on the differences in the cultural perception of the capital by users with the two different attributes.

A total of 1,427,000 eligible data points were obtained after filtering, with 639,000 data points from local residents, accounting for 44.8% of the total, and 788,000 data points from tourists, accounting for 55.2%. The percentage of perceptions of the capital's culture by users of the two attributes were calculated separately (Figure 7), and it was found that both local residents and tourists have a strong perception of the Beijing-style culture. In addition, the perceptions of ancient capital culture and red culture are stronger for tourists than for local residents, while local residents in Beijing have a higher perception of innovative culture than tourists.

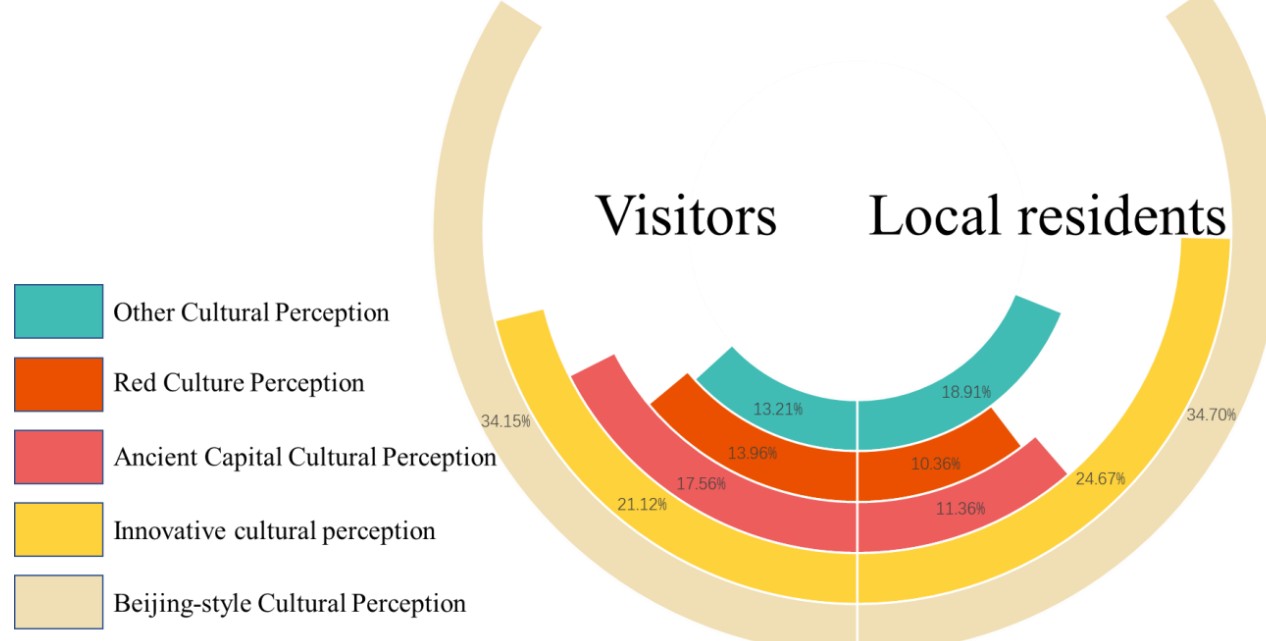

**Figure 7.** Percentage of cultural perceptions of users with different attributes.

**5. Discussion and Conclusions**

Based on ancient capital culture, red culture, Beijing-style culture, innovation culture, and other types of culture in Beijing, this study constructed a cultural perception symbol system and collected Weibo pictures. With the help of a deep learning photo recognition method, it identified and analyzed the public perception of the five types of culture in each block. Deep learning technology was introduced into the analysis of the cultural perception

of historical and cultural blocks to provide technical support and new research ideas and methods for the protection and utilization of historical and cultural blocks.

As an important part of the intangible aspects of cities, culture exists in all aspects of cities. As an urban entity inherited from history, historical and cultural blocks contain rich cultural connotations and are the focus of current research [12–14], but the research content is mostly qualitative [17] and the research methods are mostly questionnaire surveys [18]. In the era of big data, new research directions have been provided for the study of cultural heritage. The use of deep learning techniques is now a hot topic in cultural heritage research [39–42]. Owing to the richness and complexity of the meaning of photo information itself, it is difficult to analyze such "ideographic data". At present, photo data-based studies on historical and cultural blocks are limited, as photo data are mostly adopted in computer-related studies [54]. In this study, the application of photo recognition technology in deep learning to the cultural perception of historical and cultural blocks is a new direction in the field of cultural heritage research. Based on photo recognition methods, this study effectively identified five types of cultural photos in the block with a recognition accuracy of 81% and achieved good research results. From the perspective of public perception, both Beijing-style culture and the ancient capital culture with the characteristics of Beijing ancient capital have been widely recognized, and the degree of perception of an innovative culture was also high, indicating that cultural inheritance and innovation have a distinct era. Red culture perception is relatively low. It can be found that although Beijing's inner city is rich in cultural resources, due to historical reasons and objective factors of urban development, the positioning of the 28 historical and cultural blocks is different, resulting in different perceptions of the public in these blocks. This shows that photo-based cultural perception research is a good choice in this era of big data.

The analysis of data from more than one million Weibo users shows that the public perception of culture in historical and cultural blocks varies and that all the five types of culture in Beijing's capital are present in the 28 historical and cultural blocks; however, the perception of different cultures varies from one block to another. The different perceptions of culture in historical and cultural blocks are caused by different cultural resources within the blocks [43]. Culture itself is an abstract perception, and as Shakespeare is attributed, "A thousand readers have a thousand Hamlets". Culture is complex, and the different perceptions reflect its complexity, which cannot be elucidated with traditional cultural perception studies using sample questionnaires. The use of big data combined with visualization techniques for cultural perception research can visually demonstrate the complexity of culture, which is helpful for the interpretation of cultural perception and for exploring a new research direction for cultural perception research.

Through a comprehensive analysis of the perception of different cultural types in the historic and cultural blocks, it can be found that the 28 historic and cultural blocks in Beijing's old city are mainly dominated by a single culture and a mixture of composite cultures. There is no historic and cultural block with a balance of the five cultures, which reflects the outstanding characteristics of cultural diversification within the block. For single-culture-dominated blocks, such as the Xisi Beitoutiao to Batiao blocks, the characteristic cultural resources can be fully explored; for blocks dominated by composite multicultural types, while inheriting historical culture, they can also combine the characteristics of other cultures, innovate in conservation, and seek better development for the block. The study of cultural perceptions provides a good basis for the conservation and heritage use of historical and cultural blocks.

In addition to the influence of the cultural resources of the blocks, the different attributes of the perceived groups also have an impact on the outcome of perception. Different groups are constrained by their own education, perceptions, and views, which can lead to differences in cultural perceptions [26–28]. While these studies have examined the perceptual differences between different perceptual groups at the macro level, this paper narrows the research level of cultural perception from the macro level to the micro level, exploring the differences in the perceptions of tourists and local residents towards the culture of the

capital. According to the analysis of the results, the local cultural characteristics of Beijing are prominent, leading to their dominance in the minds of the general public. As for the other three types of culture, local residents will not likely pay attention to ancient capital culture and red culture, as these have long become the norm in their lives, but will pay more attention to innovative culture. Beijing, as a historical city and capital, the ancient charm of history, and the love of motherland are the main reasons that attract tourists to visit the city; therefore, tourists will have a high degree of perception of ancient capital culture and red culture. These suggest that a deeper understanding of cultural perception is necessary, which requires the support of rich multidimensional data.

However, there are some shortcomings in this study. Firstly, this study uses AI technology to categorize photos for cultural perception, but due to the complexity of culture, the content of some photos may involve multiple cultural elements at the same time. With the development of AI technology, the application of multi-labeling technology to the cultural perception study of photos would enhance the accuracy of the relevant research. Secondly, historical and cultural districts are a complex cultural construct, and quantitative analysis relying on big data alone cannot fully explore the cultural connotations of historical and cultural districts. Qualitative research methods (e.g., observational research, interview research) should be added to the study to enhance the depth of cultural connotations in historic districts. Finally, the data in this study came from social media, and we performed only a preliminarily analysis of the cultural perception differences between local residents and tourists, lacking an in-depth exploration of the cultural perception differences between these two groups. Further systematic exploration of the heterogeneity of cultural perceptions between different groups needs to be strengthened in the future.

**Author Contributions:** Conceptualization, B.M. and J.W.; Data curation, N.L., Z.Q. and J.L.; Writing—original draft, S.C. All authors have read and agreed to the published version of the manuscript.

**Funding:** This research was funded by the Institute of Beijing Studies, Beijing Union University, the National Natural Science Foundation of China (Grant Nos. 41671165,51878052), the Academic Research Projects of Beijing Union University (Grant Nos. ZK40202001, RB202101, BPHR2020DZ01), and Science and Technology General Projects of Beijing Municipal Commission of Education (Grant Nos. KM202011417013).

**Institutional Review Board Statement:** Not applicable.

**Informed Consent Statement:** All participation was voluntary and verbal consent was obtained from all subjects involved in the study.

**Data Availability Statement:** Not applicable.

**Acknowledgments:** We would like to especially thank the 4 anonymous reviewers for their patient and meticulous work and valuable revision suggestions. Special thanks to Professor Bo Zhang for her valuable comments and suggestions during the revision process of the article. Many thanks also to M.A. Bin Tian and M.A. Guoqing Zhi for their valuable comments. Thanks for the funding of "innovation project for students of advanced and sophisticated disciplines of Beijing Studies".

**Conflicts of Interest:** The authors declare no conflict of interest.

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
