# Peer review of "Cultural Perception of the Historical and Cultural Blocks of Beijing Based on Weibo Photos"

_land, doi:10.3390/land11040495_

Round 1
Reviewer 1 Report
Comments and Suggestions for Authors
Dear authors,
Congratulations for the work done in making this article. It is clear that you have made a sustained effort, but publishing an article is a process of continuous learning. As a result, please find below some remarks, on each section. I encourage you to implement each one separately, so that the article has a better quality than the current one.
Introduction
GAP of the study (Why do you implement this study? Why there is the need of this analysis in the literature?)
Theoretical background
I think that the authors should discuss more in-depth the analysis of the theoretical background. You should have a small section in the article to support this statement.
Language
Review the english language by a certified professional
References
I reccommend also to see other references as well as:
The Influence of Cross-Cultural Awareness and Tourist Experience on Authenticity, Tourist Satisfaction and Acculturation in World Cultural Heritage Sites of Korea.
MacCannell, D. The Tourist; Schocken Books: New York, NY, USA, 1976 2.
Wang, N. Rethinking authenticity in tourism experience. Ann. Tour. Res. 1999, 26, 349–370.
Author Response
请参阅附件。

Reviewer 2 Report
The article is interesting and offers an innovative methodology in the field of cultural heritage research. The research study is all encompassing, consisting of several "layers" of research.
Only minor suggestions are offered below:
Table 1. does not add very much to the text, it can be omitted.
Table 3. needs to be referred to in the text.
Materials and Methods - As much methodology is largely described, parts of it (especially Figure 2) are not self-understandable. Figure 2 aims to offer a visually attractive explanation of the study framework but visual symbols are more confusing than understandable.
155 - Error! Reference source not found
161 - Weibo Weibo
References need to be harmonized according to the journal policy.
Reviewer 3 Report
Dear Authors,
Thank you for writing this compelling manuscript. The study is supported by appropriate aims and a suitable research design and the conclusions drawn are overall logical. Nevertheless, the paper can be improved in the following respects:
(i) Provide a clearer explanation of the novel contributions of this research to theory/academic debates and methodological approaches as employed in this interdisciplinary field of inquiry.
(ii) Provide a thicker explanation of how the specific choice of methods and techniques have been informed by specific existing thematic and methods literature.
(iii) Provide a deeper critique of the limitations of the study i.e. image recognition. Relatedly, consideration can be given to how qualitative research, e.g. observational and interview-based research, could supplement this study.
(iv) The distinction that this study makes between 'local residents' and 'tourists' is potentially over-simplified and should be engaged with more critical depth in the literature review, analysis, and concluding discussion. E.g., what about former residents who have developed a deep relationship with the place but return as tourists?
(v) The readability and visual quality of the figures can be improved.
Reviewer 4 Report
The main question addressed by the research is the cultural perception the historical and cultural blocks of Beijing based on Weibo images, which is an important theme in the field of study because it proposes a specific way of studying historical and cultural blocks. This is also the added value of the research. The methodology is good and the analysis is carried out correctly with an in-depth analysis of the results. Figures and tables are ok.
Nevertheless, the weaknesses of this paper is the introduction (specifically the literature review). Please, address the following points:
- in line 59 say more about the research that followed the book "Revitalizing Historic Urban Quarters". Have some researches address the issues of this book? In which way they advance the literature?
- in the following paragraph (lines 60 to 66), please refer to more inquiries on cultural heritage, such as the paper published in sustainability "Military Barracks as Cultural Heritage in Italy: A Comparison between before-1900- and 1900-to-1950-Built Barracks" (https://www.mdpi.com/2071-1050/13/2/782) which is surely useful for framing the literature review for you. The book "Historic Cities: Issues in Urban Conservation" by The Getty Conservation Institute, Los Angeles, 2019 is also a relevant reference that collect some studies on planning of historic cities and specific blocks. these two main reference will help you to boost the internationalisation of your literature review
I am looking forward to revise your update paper.
Round 2
Reviewer 3 Report
Thank you for accommodating the recommended amendments. The revisions have been made to my satisfaction.
Reviewer 4 Report
The paper is ready for publication.